# Cognitive Function and the Ability to Operate Long-Term Oxygen Therapy Equipment: An Exploratory Study

**DOI:** 10.3390/ijerph191710708

**Published:** 2022-08-28

**Authors:** Hiroki Annaka, Tomonori Nomura, Hiroshi Moriyama

**Affiliations:** 1Department of Occupational Therapy, National Hospital Organization Nishiniigata Chuo Hospital, Niigata 950-2085, Japan; 2Graduate School, Niigata University of Health and Welfare, Niigata 950-3198, Japan; 3Department of Occupational Therapy, Faculty of Rehabilitation, Niigata University of Health and Welfare, Niigata 950-3198, Japan; 4Respiratory Center, National Hospital Organization Nishiniigata Chuo Hospital, Niigata 950-2085, Japan

**Keywords:** lung disease, long-term care, cognitive dysfunction, rehabilitation, oxygen inhalation therapy

## Abstract

Chronic respiratory disease patients with severe hypoxia receive long-term oxygen therapy (LTOT). The proper operation of LTOT equipment is essential for continuing treatment. This exploratory study investigated the relationship between cognitive impairment as a comorbidity in patients receiving LTOT and their ability to operate the LTOT equipment. The study measured responses to questions based on the ability of participants to operate the equipment and applied the Montreal Cognitive Assessment (MoCA). The ability of groups with MoCA scores ≤ 25 and >25 to operate LTOT equipment was compared to confirm the correlation between MoCA and ability to operate the equipment. An aggregate of 60 participants receiving LTOT were recruited, of whom 45 (75%) were MoCA score ≤ 25. The group of MoCA score ≤ 25 demonstrated a lower ability to operate LTOT equipment than group of MoCA score > 25 (*p* = 0.012). Additionally, a correlation was found between the ability to operate LTOT equipment and MoCA (rs = 0.743, *p* < 0.001). The results indicated that the group of MoCA score ≤ 25 indicated a lower ability to operate LTOT equipment than that of MoCA score > 25. Cognitive impairment in patients receiving LTOT can affect their ability to operate LTOT equipment.

## 1. Introduction

Long-term oxygen therapy (LTOT) allows the inhalation of oxygen at home, improving the survival rates and quality of life of patients with chronic respiratory disease (CRD) and severe hypoxia [1]. To benefit from LTOT, patients must control equipment such as concentrators and portable oxygen [2,3]. Every nation has developed guidelines for LTOT to maximize benefits for patients receiving LTOT [1,2,3]. Healthcare workers reference such guidelines to assist patients in adhering appropriately to LTOT regulations [4,5].

LTOT is advantageous for patients but also poses the potential risk of adverse events [1,3,6,7,8] such as explosions or fires during oxygen inhalation, the exacerbation of respiratory disease because of too much or too little oxygen inhalation, or falls because of contact with the equipment or cannula, which can lead to hospitalizations and even mortality [3,6,7,8,9]. Such occurrences are almost always triggered by hazardous behaviors performed by patients receiving LTOT. Healthcare workers have conducted patient education to inhibit risky conduct; however, adverse events continue to be reported [1,8]. Previous studies have reported that hazardous behaviors indicate human errors in the operation of LTOT equipment [2,3,8,9]. Hence, healthcare workers must confirm that patients properly operate LTOT equipment.

Cognitive impairment is a comorbidity related to CRD [10] and is presented in 36–77% of patients receiving LTOT [11,12,13]. Patients with cognitive impairment generally confront difficulties in operating everyday technology such as remote controls, washing machines, and telephones [14,15,16]. It has been reported that cognitive function and equipment operations are also similarly correlated in CRD patients. Cognitive impairment causes errors in the use of inhalers used to prevent exacerbations for CRD patients [17]. The existing studies have further reported that CRD patients with cognitive impairment cannot master inhaler operations despite the delivery of patient education by healthcare workers [18]. Cognitive impairment interventions may be required to assist patients in appropriately using inhalers [19,20]. Patients receiving LTOT also require cognitive function to properly operate LTOT equipment. We speculate that patients with cognitive impairment cause operational errors in the use of LTOT equipment. To our knowledge, no study has yet devised a questionnaire to investigate the operation of LTOT equipment. It also remains unclear whether cognitive impairment affects the utilization of LTOT equipment.

This exploratory study aims to purposed to (1) draft a survey question to measure the ability to operate LTOT equipment and (2) investigate the relationship between cognitive function and the ability to operate LTOT equipment in people with CRD.

## 2. Materials and Methods

### 2.1. Study Design

This exploratory study used a single-center cross-sectional design and was conduct-ed from December 2019 to April 2021. The participants used LTOT at home. We measured the cognitive function of participants and interviewed their family members regarding the ability of participants to operate the LTOT equipment. The interviews also collected demographic data about participants and their families. Medical charts were used to gather medical information about participants.

### 2.2. Participants

We enrolled participants and their family members from the outpatient department of the National Hospital Organization Nishiniigata Chuo Hospital in Niigata in Japan. The study applied the following inclusion criteria: (a) diagnosed with CRD, (b) receiving LTOT, (c) living with family, and (d) native Japanese speaker. Participants were excluded if they had (a) a history of neurologic brain disease, (b) a psychiatric disorder, (c) an upper limb disorder that could influence LTOT operations, or (d) relevant depressive symptoms as determined by a Hamilton Depression Rating Scale score of ≥ 8 [21].

The study was conducted in accordance with the Helsinki Declaration and the research protocol was approved by the ethics committee of Nishi-Niigata Chuo Hospital (approval number 1921) and Niigata University of Health and Welfare (approval number 18387–200313). All the participants provided written informed consent before the study was conducted.

### 2.3. Measurement

We interviewed participants to ascertain their characteristics (age, sex, education) and their family attributes (age, sex, relationship with participants), and collected the fol-lowing medical information from their medical records: disease, spirometry (forced expir-atory volume 1.0% and %vital capacity), modified Medical Research Council Dyspnea scale, and LTOT (history, oxygen flow, type of equipment). The LTOT Equipment Opera-tion Ability Survey, Montreal Cognitive Assessment (MoCA), and Lung Information Needs Questionnaire (LINQ) were administered by an occupational therapist (H.A) for this study.

#### 2.3.1. LTOT Equipment Operation Ability Survey

We drafted a survey to measure the ability to operate LTOT equipment. The question instrument referenced Japanese guidelines regarding oxygen therapy and related to ad-verse events at home (Table 1) [2]. The question generation was discussed by multiple respiratory specialists with extensive clinical experience. We drafted a 5-question instrument to measure the actual utilization of the equipment, assuming that errors in the operation of the selected equipment were associated with hazardous behaviors causing adverse events (Table 1). Responses to each item were plotted on a 7-point Likert scale (1: “full assistance” and 7: “completely independent”), and the total scores of five questions (minimum 5 points, maximum 35 points) were deemed the outcome. Table 2 presents the survey used in this study. Scores closer to 35 points indicated the correct operation of the equipment (Table 2). The survey was piloted to ensure that the questions were easy for the participants’ family members to read and understand. The self-administered survey was presented on paper to the family members of participants, who scored the status of correctly operating the equipment at home on a 7-point scale (Table 2). After the validity and reliability were confirmed, the survey was used to ascertain the relationship between participants’ cognitive function and their ability to appropriately utilize LTOT equipment.

#### 2.3.2. Lung Information Needs Questionnaire (LINQ)

A psychological scale, LINQ assesses the self-management knowledge of patients with CRD [22]. This questionnaire encompasses six domains: disease knowledge, medication, self-management, smoking, exercise, and nutrition. It yields scores between a minimum of 0 and a maximum of 25. We confirmed criterion-related validity by examining the correlations of the self-administered questionnaire to LINQ. This study obtained objective answers by performing LINQ assessments for the participants’ family members.

#### 2.3.3. Montreal Cognitive Assessment (MoCA)

Cognitive function was measured through MoCA, a screening test for mild cognitive impairment [23]. MoCA is commonly used for CRD [12,24]. This study was classified into two groups based on the MoCA cut-off score of ≤25 [23].

### 2.4. Statistical Analysis

Statistical analyses were performed using IBM SPSS Statistics for Windows, Version 20.0 (IBM Japan, Tokyo, Japan).

First, we confirmed the validity and reliability of the survey on the ability to operate LTOT equipment. The construct validity was assessed based on exploratory factor analysis (EFA). The number of factors was determined on the basis of a scree plot [25]. The analysis used the least-squares method with the ProMax rotation. Items were excluded from the questionnaire if their factor loadings were computed below the predefined cut-off value of 0.7 [25]. Criterion-related validity was confirmed by examining the correlations between the question and the LINQ. Reliability was determined through Cronbach’s alpha coefficient [26].

Second, we verified the relationship between the ability to operate LTOT equipment and cognitive function. The Shapiro–Wilk test was applied to verify the distribution of the ability to operate LTOT equipment survey and MoCA. We classified participants into two groups, those with a MoCA score > 25 and those with a MoCA score ≤ 25 by referencing their MoCA scores. Then, we used the Mann–Whitney U test to compare the differences between the groups in the ability to operate LTOT equipment survey. In addition, Spearman’s rank correlation analysis was applied to ascertain the relationship between the ability to operate LTOT equipment operation survey and MoCA. All analyses were performed using a two-sided test, and statistical significance was set to *p* < 0.05.

The sample size was determined via the Mann–Whitney U test and EFA. The Mann–Whitney U test sample size was calculated using G * Power version 3.1.9.4; Heinrich-Heine-Universität Düsseldorf, Düsseldorf, Germany. Test family: *t*-test, statistical test: Wilcoxon–Mann–Whitney test (two groups), tails: two, parent distribution: Laplace, Effect size: d 0.8, the Mann–Whitney U test and allocation ratio: 3. These computations were elucidated as 12 MoCA score > 25 and 34 MoCA score ≤ 25. Additionally, the sample size for EFA required 10 or more participants per question or 50 or more people [25]. Thus, the sample size for this study was set to 50 or more, with 15 or more participants for MoCA score > 25 and 35 or more participants for MoCA score ≤ 25.

## 3. Results

### 3.1. Characteristics of Patients and Their Family Members

Table 3 displays the characteristics of participants and their family members. The MoCA score denoted a median of 23.0 (18.0–25.7); 15 participants (25%) were considered in the MoCA score > 25, and 45 (75%) were categorized in the MoCA score ≤ 25 cohort.

### 3.2. Validity and Reliability of the Survey on the Ability to Operate LTOT Equipment

Initially, we confirmed the construct validity of the question. The scree plot on survey evaluating the ability to operate LTOT equipment demonstrated one factor. The EFA evidenced a factor loading of 0.7 or more for each question (Table 4). Second, we confirmed the criterion-related validity. The results of the assessment of criterion-related validity evinced significant and suitable correlations between the LTOT equipment operation question and the LINQ for the total score (rs = −0.492, *p* < 0.01). Finally, we confirmed the reliability of the question, whose Cronbach’s alpha coefficient was calculated at 0.932 for the total score (Table 4).

### 3.3. Relationship between LTOT Equipment Operation Ability Survey and MoCA

The MoCA scores for the group of MoCA score > 25 denoted a median of 26.0 (26.0–28.0) and a median of 21.0 (17.0–24.0) was computed for the group of MoCA score. Hence, the two groups evidenced a significant difference (*p* < 0.001). The total scores of the survey on the ability to operate LTOT equipment yielded a median of 35.0 (35.0−35.0) for the group of MoCA scores > 25 and a median of 34.0 (28.0–35.0) for the group of MoCA scores ≤ 25; a significant difference was thus evidenced between the groups (*p* = 0.012) (Figure 1). The ability to operate LTOT equipment survey and MoCA also demonstrated a correlation (r = 0.743, *p* < 0.001) (Figure 2).

## 4. Discussion

This study developed a survey on the ability to operate LTOT equipment. It also investigated the relationship between such capability and cognitive functions. Participants with MoCA scores ≤ 25 displayed a lower ability to operate LTOT equipment vis-à-vis participants with MoCA scores > 25.

This study designed and drafted a survey focusing on the ability to operate LTOT equipment. We had not been able to find a survey questionnaire measuring the ability to appropriately utilize LTOT equipment in the extant existing literature. The implications of measuring such a capability could confirm whether patients appropriately inhale oxygen [3,8] and help prevent equipment operation errors because of hazardous behaviors by patients [8]. Healthcare workers engaged in regular outpatient care find it difficult to decide whether patients are appropriately operating the LTOT equipment at home. The apprehension of a patient’s ability to properly use the LTOT equipment is one way to check the safe use of LTOT equipment by patients. Therefore, healthcare workers must grasp the ability of patients to aptly utilize LTOT equipment for the continuation of their LTOT.

This study investigated the relationship between cognitive function and the ability of patients to appropriately use LTOT equipment. It found that patients receiving LTOT MoCA score ≤ 25 demonstrated poor capability to operate LTOT equipment. Hypoxia-related changes in brain structure cause cognitive impairments in patients receiving LTOT [10]. These deficiencies become worse when CRD is accompanied by severe hypoxia [27,28]. Patients with cognitive impairment find it difficult to understand how to operate equipment like LTOT, such as dealing with equipment functions or pushing buttons properly [14]. CRD patients with cognitive impairment could find it problematic to recall and execute new information on which they have been instructed [19,20]. Cognitive impairment related to poor ability to operate LTOT can result in hazardous behaviors. Further, the continuing use of LTOT for such patients could require the assistance of equipment operators. Hence, cognitive impairment can affect the independence and quality of life of patients receiving LTOT.

Therefore, it must be ensured that patients appropriately operate the relevant equipment to continue LTOT. We suggest that healthcare workers must detect the early signs of cognitive impairment in patients receiving LTOT and must apprehend the accuracy of the utilization of LTOT equipment.

We must acknowledge several limitations of the present study. The results of this study evidenced a correlation between cognitive function and the ability of patients to appropriately utilize LTOT equipment. However, a causal relationship was not established because this study was cross-sectional. Nevertheless, to our knowledge, our study is the first to investigate the ability of patients to operate LTOT equipment. Therefore, this result could represent valuable exploratory data. In addition, this study drafted a trial survey on the ability to operate LTOT equipment but referenced Japanese guidelines only. The study’s focus on hazardous behaviors vis-à-vis equipment operations is commonly reported, although equipment types and support systems differ in each country. Future studies will consider the development of usable questionnaires for each country in consideration of the questionnaire developed herein. Finally, the survey in this study may not fully measure the equipment operation process. Developing a questionnaire for LTOT equipment operation ability requires a process of carefully selecting from a larger number of hypothetical question items and a suitable sample size [29]. We aim to complete the questionnaire by reexamining the question content in collaboration with more experts and conducting an investigation with a larger sample size.

## 5. Conclusions

We explored the relationship between the ability to operate LTOT equipment and the cognitive function of patients receiving LTOT. The study outcomes indicated that patients MoCA score ≤ 25 exhibited a lower ability to operate LTOT equipment than patients MoCA score > 25. Cognitive impairment in patients receiving LTOT can hence affect the ability to appropriately utilize LTOT equipment.

## Figures and Tables

**Figure 1 ijerph-19-10708-f001:**
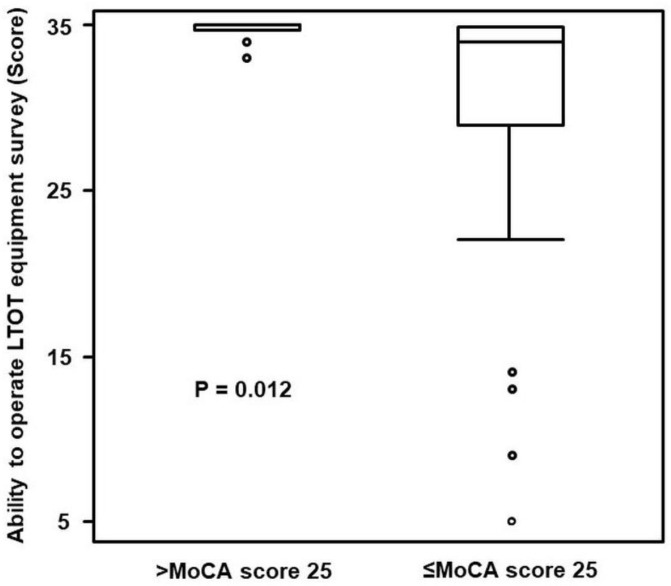
Comparison of Montreal Cognitive Assessment (MoCA) score ≤ 25 or >25 and the ability to operate long-term oxygen therapy equipment. The group of MoCA score ≤ 25 exhibited a lower ability to operate long-term oxygen therapy (LTOT) equipment survey than the group of MoCA score > 25 (*p* = 0.012).

**Figure 2 ijerph-19-10708-f002:**
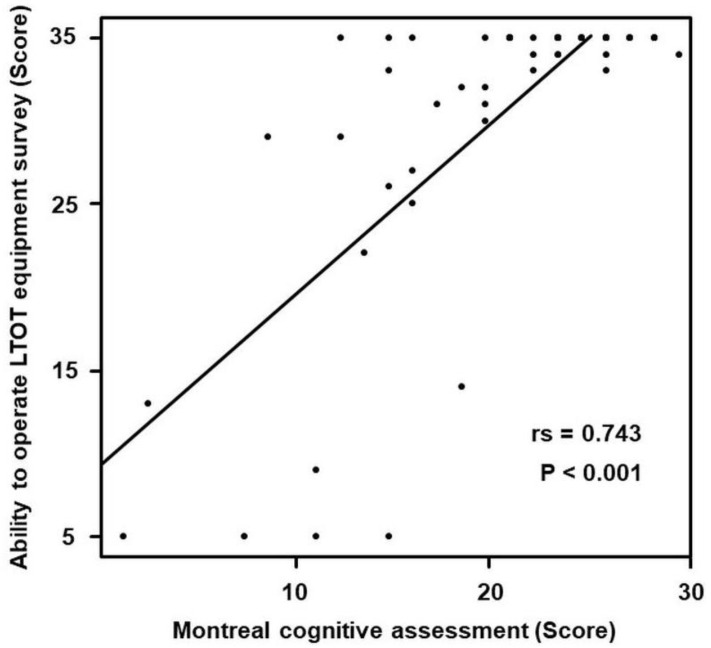
Correlation between the ability to operate long-term oxygen therapy (LTOT) equipment survey and Montreal Cognitive Assessment. The ability to operate LTOT equipment and cognitive function (rs = 0.743, *p* < 0.001) were positively correlated.

**Table 1 ijerph-19-10708-t001:** Hazardous behaviors and adverse events related to the ability to operate long-term oxygen therapy equipment and questions to measure them.

Hazardous Behaviors	Adverse Events	Equipment Operation	Question
Smoking/contact with open flames	Explosion and fire	Avoidance of fire	Can you avoid causing fires while using the home oxygen equipment?
Poorly adjusted flow rate	Worsening respiratory failure	Adjustment of flow rates	Can you adjust the flow rate while in the toilet, in the bath, or going out?
Lack of understanding about portable oxygen	Running out of oxygen	Operation of portable oxygen	Can you use portable oxygen?
Stumbling on the equipment and portable oxygen	Falls	Avoidance of contact with the cannula and portable oxygen	Can you avoid contact with the cannula and portable oxygen?
Lack of understanding about the equipment	No oxygen supply	Dealing with alarms	Can you deal with alarms regarding forgetting to shut the cylinder or running out of oxygen?

**Table 2 ijerph-19-10708-t002:** The survey of long-term oxygen therapy equipment operation ability drafted for this study.

Question 1: Can you avoid causing fires while using the home oxygen equipment?								
Full assistance	①	②	③	④	⑤	⑥	⑦	Completely independent
Question 2: Can you adjust the flow rate while in the toilet, in the bath, or going out?								
Full assistance	①	②	③	④	⑤	⑥	⑦	Completely independent
Question 3: Can you use portable oxygen?								
Full assistance	①	②	③	④	⑤	⑥	⑦	Completely independent
Question 4: Can you avoid contact with the cannula and portable oxygen?								
Full assistance	①	②	③	④	⑤	⑥	⑦	Completely independent
Question 5: Can you deal with alarms regarding forgetting to shut the cylinder or running out of oxygen?								
Full assistance	①	②	③	④	⑤	⑥	⑦	Completely independent

**Table 3 ijerph-19-10708-t003:** Characteristics of the participants and their family members.

Characteristics	Participants (*n* = 60)	Families (*n* = 60)
Age (years)	77 (6) *	70 (11) *
Gender		
Male	48 (80%) †	11 (18%) †
Female	12 (20%) †	49 (82%) †
Education (years)	12.0 (9.0–12.0) ‡	
Relationship with patients		
Spouse		36 (65%) †
Son/daughter		16 (27%) †
Other		8 (8%) †
Disease		
Chronic obstructive pulmonary disease	24 (40%) †	
Interstitial pneumonia	23 (38%) †	
Other	13 (22%) †	
Spirometry		
Forced expiratory volume 1.0%	74.1 (50.0–89.5) ‡	
Group of COPD ^1^	47.0 (39.0−53.5) ‡	
Group of interstitial pneumonia	89.6 (82.9−94.4) ‡	
Group of other diseases	71.9 (60.0−85.2) ‡	
%Vital capacity	66.4 (57.8–84.8) ‡	
Group of COPD ^1^	72.4 (63.3−96.5) ‡	
Group of interstitial pneumonia	66.0 (59.0−82.4) ‡	
Group of other diseases	56.3 (40.9−69.5) ‡	
mMRC ^2^	2.0 (1.0–3.0) ‡	
LTOT ^3^		
History (months)	13.0 (7.0–41.5) ‡	
Oxygen flow (L)	3.0 (2.0–4.0) ‡	
Concentrator	56 (92%) †	
Liquid oxygen	4 (8%) †	
Portable oxygen	60 (100%) †	
Lung Information Needs Questionnaire	2.0 (1.0−4.75) ‡	
LTOT Equipment Operation Ability Survey	35.0 (31.0−35.0) ‡	
Montreal cognitive assessment	23.0 (18.0–25.7) ‡	
score ≤ 25	45 (75%) †	

*: mean (standard deviation), †: *n* (%), ‡: median (interquartile range). ^1^ COPD = Chronic obstructive pulmonary disease, ^2^ mMRC = modified Medical Research Council Dyspnea scale, ^3^ LTOT = long-term oxygen therapy.

**Table 4 ijerph-19-10708-t004:** Exploratory factor analysis and Cronbach’s coefficient alpha: survey regarding the ability to operate long-term oxygen therapy equipment (*n* = 60).

Survey	Factor Loading	Cronbach’s Coefficient Alpha
Total score of LTOT ^1^ equipment operation ability survey (questions 1−5)	−	0.932
Question 1 Can you avoid causing fires during the use of the home oxygen equipment?	**0.911**	0.908
Question 2 Can you adjust the flow rate in the toilet, in the bath, or going out?	**0.824**	0.921
Question 3 Can you use portable oxygen?	**0.840**	0.923
Question 4 Can you avoid contact with the cannula and portable oxygen?	**0.894**	0.910
Question 5 Can you deal with alarms regarding forgetting to shut the cylinder or running out of oxygen?	**0.836**	0.919

Loadings > 0.7 are indicated in bold. Least-Squares method, ProMax rotation. ^1^ LTOT = long-term oxygen therapy.

## Data Availability

The data presented in this study are available in Appendix A.

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
