# Peer review of "Cognitive Function and the Ability to Operate Long-Term Oxygen Therapy Equipment: An Exploratory Study"

_ijerph, 2022, doi:10.3390/ijerph191710708_

Round 1

Reviewer 1 Report

Thank you for your hard work in producing this manuscript. This is a very interesting topic and your results are valuable to health professionals

To improve the flow of the manuscript, I have made some suggestions below.

Abstract

ü  The abstract is clear and concise with the aims well set out

ü  Page 1. line 24: ‘evinced’: I am not sure what this word means?

Introduction

ü  Page 2, line 53: ‘extant’: not sure what you mean here?

ü  Page 2, line 56: ‘Patients receiving LTOT also require cognitive functions to properly operate LTOT equipment’. Perhaps this should read: ‘Patients receiving LTOT require normal cognitive function to properly operate LTOT equipment’. No s on function

ü  Rewrite the aims as: ‘This exploratory study aims to purposed to 1) draft a survey question to measure the ability to operate LTOT equipment and 2) investigate the relationship between cognitive functions and the ability to operate LTOT equipment in people with chronic respiratory disease’.

Methods

Study design

ü  I would recommend using the word ‘participants’ rather than patients. They are your patients but when they agree to be in a study then they become participants.

ü  Page 2, line 68: in the study design – tell us the type of study and type of participants.

Patients / Participants

Page 2, line 79: the sentence is incomplete

Measurements

ü  List all measurement here including the Survey about use of LTOT, MoCa, LINQ

ü  2.3.1: add survey to the end of this heading: ‘LTOT Equipment Operation Ability Survey’ and refer to the questionnaire as a survey throughout the manuscript

ü  Line 87: use survey instead of question

ü  Line 87: use questions instead of queries

ü  Line 94: ‘A pilot study confirmed that the text of the questions was framed in a manner that was easy for the family members of patients to understand’ – perhaps better to write ‘the survey was piloted to ensure that the questions were easy for the patients’ family members to read and understand’

ü  Line 96: The self-administered survey

ü  About the survey: you will need to describe the survey in more detail e.g., how many questions? Were they all Likert scale answers or were some open questions?

ü  I think it would be a good idea to include the survey in the manuscript not just the answers

ü  Line 97: Rewrite as: ‘After the validity and reliability were confirmed, the survey was used to ascertain the relationship between participants’ cognitive function and their ability to appropriately utilize LTOT equipment’.

ü  Line 110: Rewrite as: for the participants’ family members of patients

ü  Line 18: change question to survey

ü  Perhaps refer to Table 1 when you are discussing the survey in this section

Statistical analysis

ü  Line 127: when you say Cognitive Function – there is no s at the end. Check throughout the manuscript

Results

ü  Table 2: Patients age: 77.5 (6.7)* you do not need to have decimals here. Make it 78 (7)

ü  Spirometry results: I think it might be better to give the mean spirometry results for the COPD group and separately for the IP group because combining them makes the spirometry look better that it probably is.

ü  No results for LINQ were given and results could be clearer and better presented. You need a Table with all main results

ü  Combine Tables 3 and 4

ü  Line 166: add survey after equipment

ü  Figure 1: add survey after equipment to the Y axis

ü  Figure 2: although you have a p value for this correlation – the plot does not really look like a good correlation. What do you think?

Discussion

ü  You have summarised the results very well in the first paragraph. Again, refer to the questionnaire as a survey

ü  Line 189: Rewrite as ‘This study designed drafted a question survey focusing on the ability to operate LTOT equipment. We had have not been able to find a survey questionnaire measuring the ability to appropriately utilize LTOT equipment in the extant existing literature

ü  Line 209: change continuance to continuing use

Thank you again for your hard work. The results are very worthwhile and as you have said are good foundations for further research

Reviewer 2 Report

This study aims to explore the relationship between the ability to operate LTOT equipment and the cognitive functions of patients receiving LTOT by using a cross-sectional design method. The research design is appropriate and clear, the statistical analysis method is reasonable. This study was meaningful for realizing the equipment operating risk of CRD patients and could be of interest to the audience of the journal. I have some comments:

1. A questionnaire with 5 questions was design to test the patients’ LTOT equipment operation ability. The questions are very important to the results of this study. The author should make clear what the principle to set up the questions is? Are 5 questions enough to measure the patients equipment operation ability? Could these questions reflect the patients’ true condition? Such as the question 3: “Can you use portable oxy-gen?” might be an ambiguity question because “can use” and “can use properly” could reflect different conditions.

2. The sample size is 60 (45 MoCA<25), is it enough?

3. Line 79 was incomplete.

4. Line 170-171, the sentence may have mistake.
